# Endometrial congestion is the only hysteroscopic finding indicative of chronic endometritis

**Mayuko Furui**[1,2,3], **Ayumu Ito**[1,2,3]*, **Yusuke Fukuda**[1,2,3], **Mami Sekiguchi**[1,2,3], **Kentaro Nakaoka**[1,2,3], **Yuko Hayashi**[1,2,3], **Yuko Tamaki**[1,2,3], **Yukiko Katagiri**[1,2,3], **Koichi Nagao**[3,4], **Masahiko Nakata**[1,2]

1 Faculty of Medicine, Department of Obstetrics and Gynecology, Toho University, Omori-nishi, Ota-ku, Tokyo, Japan, 2 Department of Obstetrics and Gynecology, Toho University Omori Medical Center, Omori-nishi, Ota-ku, Tokyo, Japan, 3 Reproduction Center, Toho University Omori Medical Center, Omori-nishi, Ota-ku, Tokyo, Japan, 4 Department of Urology, Toho University Omori Medical Center, Omori-nishi, Ota-ku, Tokyo, Japan

* ayumu.itou@med.toho-u.ac.jp

**Data Availability Statement:** All data generated or analyzed during this study are included in this published article and its supplementary information files.

## Abstract

Chronic endometritis (CE), an inflammatory condition characterized by plasma cell infiltration within the endometrial stroma, is prevalent among women experiencing unexplained infertility or recurrent miscarriages. CE is traditionally diagnosed by endometrial biopsy using CD138 immunohistochemistry staining. Despite some studies suggesting hysteroscopy as an alternative diagnostic tool, its reliability compared with biopsy remains controversial. This study evaluated the diagnostic accuracy of hysteroscopy for CE by examining endometrial features, such as congestion, micropolyps, edema, and polyps, and comparing these with biopsy-confirmed cases of CE. This retrospective observational study was conducted at Toho University Omori Medical Center between June 2017 and November 2019 and included patients undergoing both hysteroscopy and histopathological examination. Endometrial congestion was identified as the only hysteroscopic finding significantly associated with CE, showing a moderate diagnostic agreement with biopsy results. These findings highlight the importance of further investigating hysteroscopic features of CE and their diagnostic implications and identify endometrial congestion as a potential predictive marker for CE.

## Introduction

Chronic endometritis (CE) is a localized inflammatory disease characterized histologically by stromal edema in the mucosal surface layer, increased stromal density due to leukocyte infiltration into the glands and stroma, and presence of plasma cells within the stroma [1]. The cause of CE has traditionally been attributed to chronic endometrial inflammation resulting from bacterial infection [2]. However, in recent years, non-infectious chronic endometrial inflammation has also been considered a possible cause [3]. CE is found in 55.7% of women

**Funding:** The author(s) received no specific funding for this work.

**Competing interests:** The authors have declared that no competing interests exist.

with unexplained infertility and 9.3%–60% of women who experience recurrent miscarriages; associations between CE and these conditions have been reported [4–8]. Traditionally, the diagnosis of CE was based on histopathological evidence of plasma cell infiltration into the endometrial stroma [9]. CD138 immunohistochemistry has recently been reported to be superior to classical hematoxylin and eosin (HE) staining in detecting the presence of plasma cells within the endometrial stroma, and it is the current gold standard for CE diagnosis [10,11]. However, there are no internationally accepted fixed diagnostic criteria for CE. In clinical practice, the diagnostic criteria used in a previous study were selected arbitrarily [12].

The use of hysteroscopy for CE diagnosis has been reported [13,14]. However, its diagnostic value for CE has not been established, and the use of hysteroscopy as a substitute for endometrial biopsy is currently not recommended [15–17]. Previous studies have reported endometrial hysteroscopic findings suggestive of CE, such as endometrial congestion, micropolyps, endometrial edema, and endometrial polyps [15–19]. Although some studies have investigated the associations between these findings and the histopathological changes in CE confirmed by biopsy, few studies have compared these hysteroscopic features with those of patients without CE. Therefore, whether these findings are suggestive of CE remains unclear.

In this study, we aimed to identify the true findings suggestive of CE on hysteroscopy by comparing cases diagnosed as CE on endometrial biopsy with those assessed as non-CE on hysteroscopy.

## Materials and methods

### Study design and data

This retrospective observational study included patients with infertility who underwent hysteroscopy and endometrial histopathological examination at the Reproduction Center of Toho University Omori Medical Center between June 2017 and November 2019. Patients who only underwent one of either hysteroscopy or endometrial histopathological examination were excluded. We collected data on these patients regarding from infertility from the electrnic medical recoads system of Toho University Omori Medical Center between August 24, 2021, and March 31, 2022. This data collection aimed to supplement valuable information on the initial patient group and to deepen our understanding of the impacts of interventions in infertility treatments.

### Hysteroscopy procedure and definition of hysteroscopy results

Hysteroscopy was performed in an outpatient setting without anesthesia using a flexible hysteroscope (Olympus HYF Type V, Tokyo, Japan) during the follicular phase of the menstrual cycle. The uterine lumen was dilated with a saline solution at a pressure of 100 mmHg and observed using the hysteroscope. The following four findings were observed: endometrial congestion, micropolyps, endometrial edema, and endometrial polyps (Fig 1). The mucous membrane of the uterine body was defined as follows: endometrial congestion, if accompanied by a reddened area localized to or scattered in the entire lumen; micropolyps, if accompanied by a mass protruding into the mucous membrane of the uterine body with a diameter of <1 mm; endometrial edema, if accompanied by an irregularly thickened endometrium with multiple microscopic pale bulges on its surface; and endometrial polyps, if accompanied by a mass sized >1 mm protruding from the endometrium. In our institution, six physicians performed hysteroscopy, with each performing 3–4 procedures per month. All six physicians were obstetricians/gynecologists who were specialists in assisted reproductive medicine, with 3–10 years of experience in the field. Chlamydial antibody testing was performed before hysteroscopy, and if a positive result was obtained, the test was repeated after the completion of antimicrobial

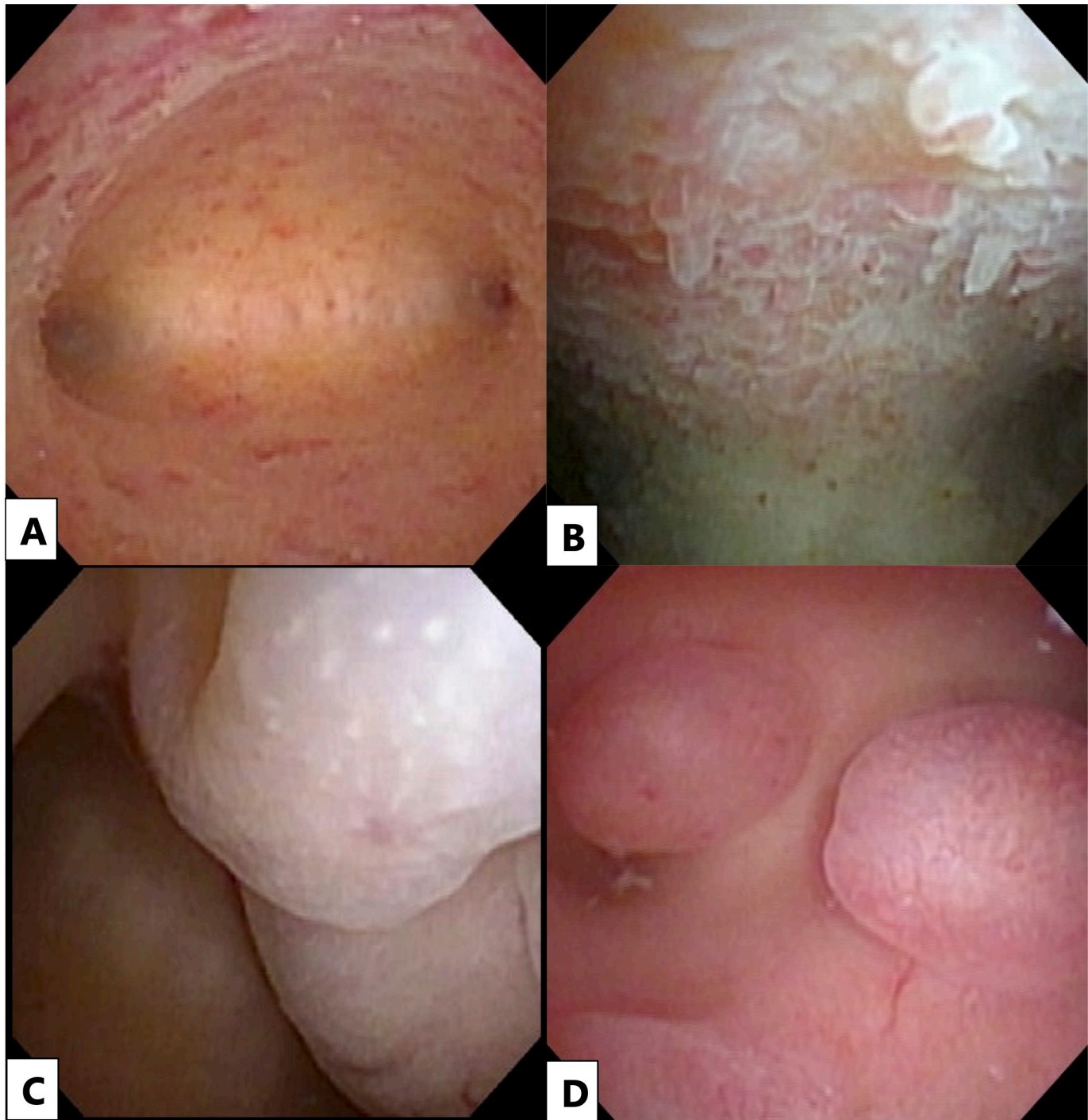

**Fig 1. Hysteroscopic findings of the uterine lumen.** (**a**) Endometrial congestion: Reddened areas localized to the endometrium of the uterine body or scattered throughout the uterine cavity. (**b**) Micropolyp: A mass sized <1 mm protruding into the endometrium of the uterine body. (**c**) Endometrial edema: Irregularly thickened endometrium with multiple minute pale bulges on its surface. (**a**) Endometrial polyp: A mass sized >1 mm protruding from the endometrium of the uterine body.

treatment. Patients were diagnosed as hysteroscopy-positive when one or more of the following features were detected: endometrial congestion, micropolyps, endometrial edema, or endometrial polyps. Patients were diagnosed as hysteroscopy-negative when none of these features were found.

## Diagnosis of CE

A definitive diagnosis of CE was established based on endometrial histopathology. Endometrial biopsy was performed using aspiration with a pipette curette (Pipet CuretTM; Cooper Surgical, Tokyo, Japan) immediately after the completion of hysteroscopy. Aspiration was performed blindly on the anterior wall of the uterus, regardless of the hysteroscopic findings. All endometrial tissue biopsy specimens were examined and diagnosed by a pathologist in the hospital pathology department. Biopsy specimens were fixed in formalin solution, and immunohistochemical staining was performed using an anti-syndecan-1 (CD138) monoclonal antibody (Monoclonal Mouse Anti-Human CD138 Clone MI15; Agilent, Tokyo, Japan). If more than five CD138 immunohistochemical staining-positive plasma cells per 20 high-power fields were detected in the endometrial tissue sample, the sample was defined as CE (CE group), and if less than five, the sample was defined as normal (non-CE group).

## Outcomes

The positive findings on hysteroscopy in the CE and non-CE groups were compared. Multivariate analysis was performed using each hysteroscopic finding as an explanatory variable and a CE histopathological diagnosis on endometrial biopsy as the objective variable, to examine the contribution of each hysteroscopic finding to the CE diagnosis. The percentage of patients diagnosed with CE/non-CE was calculated for each hysteroscopic finding to examine the diagnostic concordance rates between the hysteroscopic findings and endometrial histopathology.

## Statistical analysis

Statistical analyses were performed using SPSS Statistics ver. 26 (SPSS Inc., Chicago, IL, USA). Data were analyzed using t-tests for parametric data and the Mann—Whitney U test for non-parametric data. Chi-square and Fisher's exact tests were performed to compare the ratios. Logistic regression analysis was used for multivariate analysis to estimate odds ratios and 95% confidence intervals (CIs). All statistical analyses were two-tailed, with a significance level of 0.05.

## Ethical considerations

This study was approved by the Ethics Committee of Toho University Medical Center Omori Hospital (approval number: M21131) on August 24, 2021. Information about the study was disclosed in an opt-out format on the hospital website, ensuring that the participants had the right to refuse participation; therefore, the need for written informed consent was waived by the Ethics Committee of Toho University Omori Medical Center. The study was conducted in a manner that patients would not be disadvantaged by non-participation. All data were anonymized. All procedures performed in this study were in accordance with the ethical standards of the institution and with the principles of the 1964 Helsinki Declaration and its later amendments. We reported our results according to the Strengthening of Reporting in Observational Studies in Epidemiology Statement.

## Results

Table 1 presents the backgrounds of the eligible patients and the indications for hysteroscopy and/or assisted reproductive technology (ART). Among the 317 eligible cases, 481 hysteroscopies were performed. Of these, 82 procedures were excluded because of indeterminate findings and other reasons. Finally, 399 eligible procedures were included in this study with 200 patients in the CE group and 199 in the non-CE group.

The prevalence of the following hysteroscopic features did not differ significantly between the CE and non-CE groups: micropolyps (36.0% vs. 31.7%, p = 0.36), endometrial edema (9.5% vs. 7.5%, p = 0.48), and endometrial polyps (19.5% vs. 21.6%, p = 0.60). Only the prevalence of endometrial congestions differed significantly between the two groups (26.0% vs. 17.1%, p < 0.05) (Fig 2).

The odds ratio (95% CI) for CE compared with non-endometrial congestion findings was 1.72 (1.05–2.82) for endometrial congestion, 1.16 (0.76–1.77) for micropolyps, 1.41(0.69–2.89) for endometrial edema, and 0.91 (0.56–1.48) for endometrial polyps. Among these hysteroscopic findings, only endometrial congestion was a significant contributing factor to CE diagnosis on endometrial histopathological examination in the multivariate analysis (Fig 3).

The diagnostic concordance rates between each hysteroscopic finding and endometrial histopathology were 60.5%, 53.3%, 55.9%, and 47.6% for endometrial congestion, micropolyps,

**Table 1. Patient background and indications for hysteroscopy and ART.**

| Characteristics (n = 261) | | Values |
|---|---|---|
| Age (years) | | 36 (24–45) |
| BMI (kg/m$^2$) | | 21.6 (15.4–37.9) |
| Parity | | 0 (0–2) |
| Miscarriage | | 0 (0–4) |
| Infertile period (years) | | 1.42 (0–12) |
| AMH (ng/mL) | | 2.48 (0–15.2) |
| Basal FSH (mIU/mL) | | 7.1 (3.2–21.6) |
| Basal LH (mIU/mL) | | 4.7 (0–17.5) |
| Basal E2 (pg/mL) | | 36.7 (0–93.6) |
| Indications for hysteroscopy (399 cycles) | | |
| | Screening for ART | 47.1% |
| | Repeated implantation failure | 33.1% |
| | Endometrial polyp | 13.8% |
| | Non-ART cycle | 5.8% |
| Indications for ART* | | |
| | Tubal factor | 6.4% |
| | Endometritis | 1.1% |
| | Anti-sperm-antibody | 2.1% |
| | Male factor | 46.8% |
| | Unexplained infertility | 13.3% |
| | Diminished ovarian reserve | 11.7% |
| | Oncofertility | 5.9% |
| | Other | 12.7% |

*These data contain duplicates.

AMH, anti-Müllerian hormone; BMI, body mass index; E2, estradiol; FSH, follicle-stimulating hormone; LH, luteinizing hormone; P4, progesterone; ART, assisted reproductive technology.

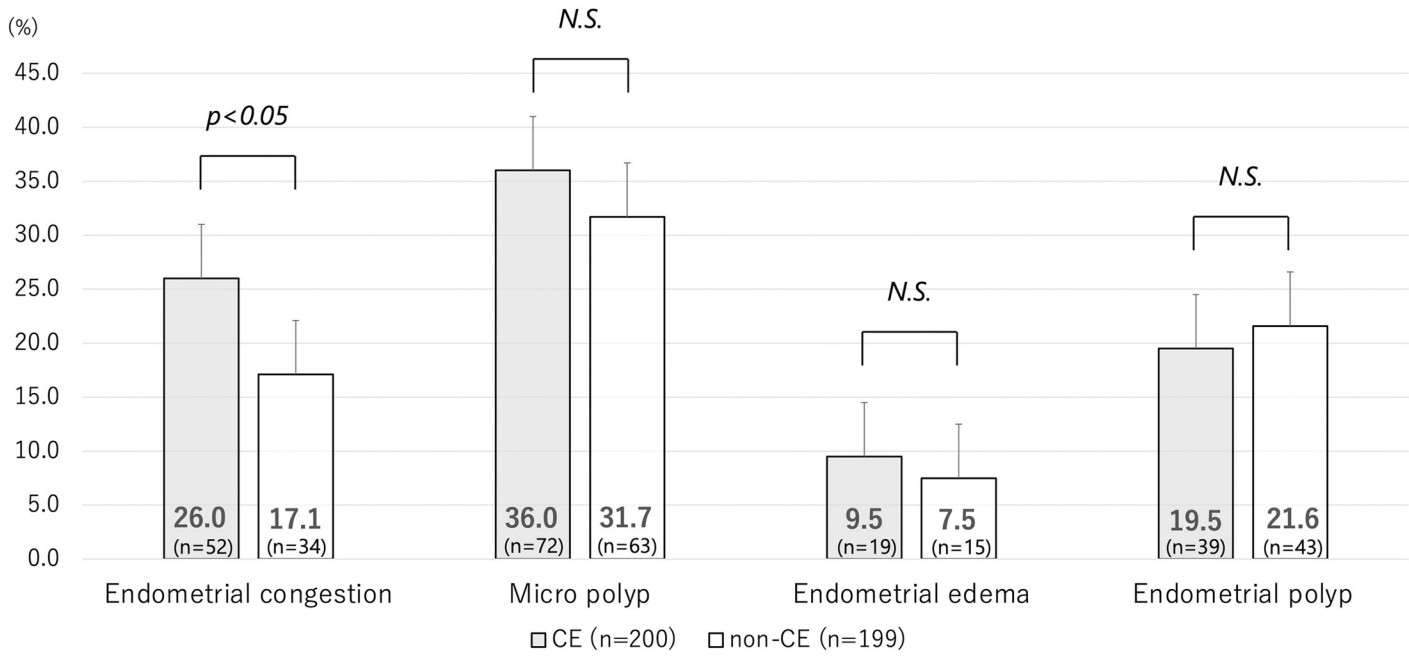

**Fig 2. Prevalence of hysteroscopic findings in CE/non-CE groups.** These data contain duplicates. CE, chronic endometritis; N.S., not significant.

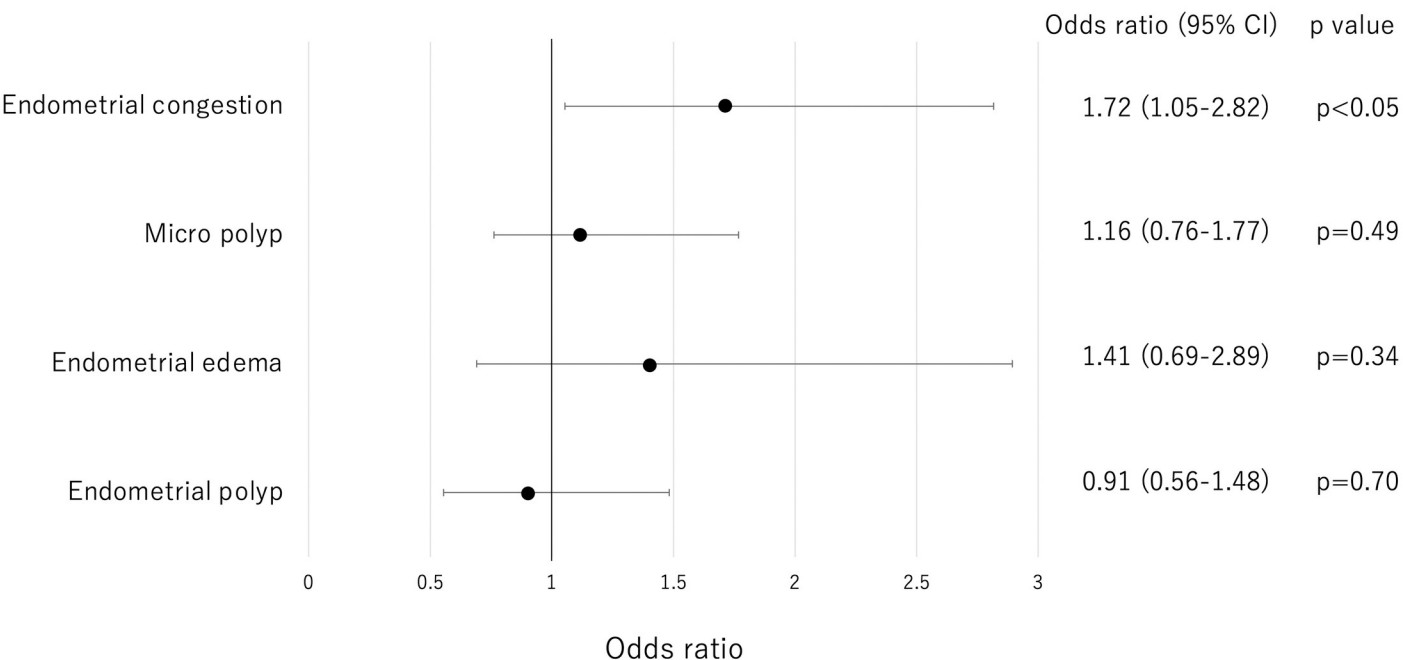

**Fig 3. Odds ratio for CE for each hysteroscopic finding.**

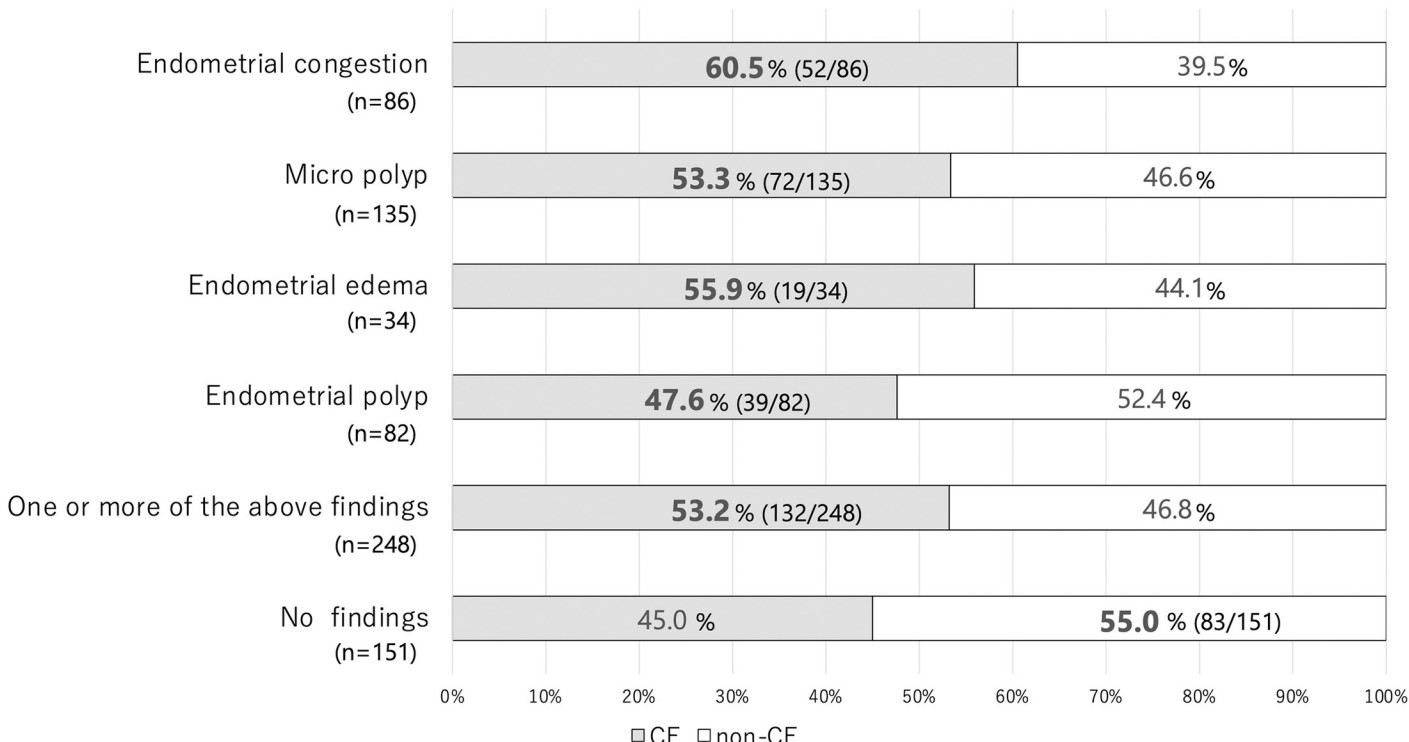

**Fig 4. Diagnostic concordance rate for CE between each hysteroscopy finding and endometrial histopathology.** CE, chronic endometritis.

endometrial edema, and endometrial polyps, respectively. The diagnostic concordance rate for CE in hysteroscopy-positive cases was 53.2%, whereas that for non-CE in hysteroscopy-negative cases was 55.0% (Fig 4).

## Discussion

The main findings of this study are as follows. First, among the hysteroscopic findings reported to be associated with CE, endometrial congestion was the only contributing factor. Second, the diagnostic concordance rate for CE between hysteroscopic findings and endometrial histopathology was low (53.0%), suggesting that the diagnosis of CE using hysteroscopy is challenging.

In the comparison of the prevalence of each hysteroscopic finding between the two groups, only the prevalence of endometrial congestion was significantly higher in the CE group than in the non-CE group. A systematic review examining the usefulness of hysteroscopy in the diagnosis of CE reported that micropolyps, endometrial edema, and diffuse or focal endometrial congestion are hysteroscopic findings suggestive of CE [18]. However, in this study, among the hysteroscopic findings suggestive of CE, only endometrial congestion had a significantly higher prevalence in the CE group than in the non-CE group, whereas other findings, such as micropolyps, endometrial edema, and endometrial polyps, were not significantly different between the two groups. Furthermore, endometrial congestion was the only contributing factor to CE diagnosis on endometrial histopathological examination in multivariate analysis. The diagnostic concordance rate for CE in patients detected as having endometrial congestion on hysteroscopy was 60.5%, whereas it was approximately 50% for other findings.

This suggests that most of the hysteroscopic findings thought to be suggestive of CE may also be present in some non-CE cases. Notably, these findings have been reported in pathophysiological conditions other than CE [20,21] which, in our view, explains the reduced diagnostic rate of CE on hysteroscopy.

The diagnostic concordance rate for CE between the hysteroscopic findings and endometrial histopathology in the present study was low (53.2%). Moreno *et al.* reported that the diagnostic discordance rate for CE between hysteroscopic findings and endometrial histopathology was high, at 58.5% (2018), which also implies a low diagnostic concordance rate [2]. A low concordance rate between hysteroscopic findings and histological examination was reported more recently, with the authors concluding that hysteroscopic signs are not yet sufficient to make an accurate diagnosis of CE [22]. The view that hysteroscopy alone cannot accurately diagnose CE is corroborated by further studies [15,18,23]. Another reason for the low diagnostic accuracy of hysteroscopy is the dependence of diagnosticians on diagnostic techniques. Additionally, endometrial histopathological examination should always be performed because the sensitivity of hysteroscopic diagnosis depends on the clinician's subjective evaluation [23]. Overall, the low diagnostic concordance rate for CE between hysteroscopic findings and endometrial histopathology can be attributed to various factors including the lack of uniform diagnostic histopathological criteria [12], low consistency in pathologists' test results [24], absence of standardized criteria for the timing of endometrial tissue collection, and reliance on the endoscopic surgeon's subjective and nonspecific observations on hysteroscopy [25]. It is important that these challenges for the diagnosis of CE are addressed in future studies.

The strength of this study is that we compared the hysteroscopic findings of CE cases with those of non-CE cases and showed that the hysteroscopic finding most suggestive of CE was endometrial congestion. However, this study has some limitations. Considering that the degree of endometrial congestion varied from considerably localized to diffuse congestion, known as the "strawberry aspect," we were not able to examine the difference in the diagnostic rate of according to differences in the degree of hyperemia. Similarly. this limitation extends to the evaluation of variations in the other findings. Another limitation is that because endometrial histopathology was performed using aspiration, the endometrial tissues were not necessarily collected from the same site as those collected using hysteroscopy. However, considering that none of these sites necessarily corresponds to the actual implantation site of the embryo, the meaningfulness of matching these sites is questionable. This study indicates that only endometrial congestion, among the hysteroscopic findings associated with previous CE, contributes to CE. However, because the diagnostic concordance rate for CE between each hysteroscopic finding and endometrial histopathology was low, we believe that hysteroscopy cannot be the first-choice diagnostic tool for CE. It is important to investigate the effect of hysteroscopic findings on clinical outcomes and the relationship between the strength of hysteroscopic findings and endometrial histopathology.

In conclusion, among the hysteroscopic findings suggestive of CE, only endometrial congestion was significantly associated with biopsy-confirmed CE. Making a diagnosis of CE based solely on hysteroscopy presents considerable challenges. Future studies should investigate the correlation between the pathological diagnosis of CE and other factors including the extent of endometrial congestion.

## Supporting information

**S1 Table. The raw data of this study.**
(XLSX)

## Author Contributions

**Conceptualization:** Ayumu Ito.

**Data curation:** Mayuko Furui, Ayumu Ito.

**Formal analysis:** Mayuko Furui, Ayumu Ito.

**Methodology:** Ayumu Ito.

**Project administration:** Ayumu Ito.

**Supervision:** Yukiko Katagiri, Koichi Nagao, Masahiko Nakata.

**Writing – original draft:** Mayuko Furui, Mami Sekiguchi.

**Writing – review & editing:** Ayumu Ito, Yusuke Fukuda, Kentaro Nakaoka, Yuko Hayashi, Yuko Tamaki, Yukiko Katagiri, Koichi Nagao.

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
