## [Decision Letter · Decision Letter 0]

4 Mar 2024

PONE-D-24-02159Endometrial congestion is the only true hysteroscopic finding suggestive of chronic endometritisPLOS ONE

Dear Dr. Ito,

Thank you for submitting your manuscript to PLOS ONE. After careful consideration, we feel that it has merit but does not fully meet PLOS ONE’s publication criteria as it currently stands. Therefore, we invite you to submit a revised version of the manuscript that addresses the points raised during the review process.

We look forward to receiving your revised manuscript.

Kind regards,

Muhammad Salman Bashir, M.S.C

Academic Editor

PLOS ONE

2. In this instance it seems there may be acceptable restrictions in place that prevent the public sharing of your minimal data. However, in line with our goal of ensuring long-term data availability to all interested researchers, PLOS’ Data Policy states that authors cannot be the sole named individuals responsible for ensuring data access (http://journals.plos.org/plosone/s/data-availability#loc-acceptable-data-sharing-methods).

Reviewers' comments:

Reviewer's Responses to Questions

**Comments to the Author**

1. Is the manuscript technically sound, and do the data support the conclusions?

Reviewer #1: Yes

Reviewer #2: Yes

2. Has the statistical analysis been performed appropriately and rigorously? 

Reviewer #1: Yes

Reviewer #2: Yes

3. Have the authors made all data underlying the findings in their manuscript fully available?

Reviewer #1: Yes

Reviewer #2: Yes

4. Is the manuscript presented in an intelligible fashion and written in standard English?

Reviewer #1: Yes

Reviewer #2: No

5. Review Comments to the Author

Reviewer #1: I read with interest manuscript title (Endometrial congestion is the only true hysteroscopic finding suggestive of chronic

endometritis), the paper provide well known information and many studies had the similar result.

It will be great to share information fron Japan sitting.

1- in demographic information many items wre missing ( number IVF trial , history of abortion, received antibiotics, ..etc).

2- the abstract need to be summarized.

Reviewer #2: The paper is very interesting. I am happy to review it.

Here there are my concerns:

- firstly, I advise you to take into account this paper to improve you reference quality with the most recent papers published on this topic (doi: 10.1007/s00404-023-07163-w;

- What could it be the solution about adhesions and their possible treatment? (doi: 10.1016/j.ajog.2021.09.015)

Line 79: the uterine cavity

Line 88: In our Institution

In general, english should be revised by mother tongue

Regarding "Hysteroscopic findings of the uterine lumen" (edit with cavity), what do you think it is more important?

I have appreciated the presence of strenghts and limitations before conclusion

Conclusions are in line with what you stated through the main text

6. PLOS authors have the option to publish the peer review history of their article (what does this mean?). If published, this will include your full peer review and any attached files.

Reviewer #1: **Yes: **Saeed Baradwan

Reviewer #2: No

---

## [Author Response · Author response to Decision Letter 0]

26 Mar 2024

March 22, 2024

Muhammad Salman Bashir, M.S.C Editor-in-Chief

Academic Editor

PLOS ONE

Dear Editor:

We are pleased to submit the revised version of our manuscript entitled, “Endometrial congestion is the only true hysteroscopic finding suggestive of chronic endometritis.” The submission ID is PONE-D-24-02159.

We thank you and the reviewers for your helpful comments and feedback. The manuscript has benefited from these insightful suggestions. The manuscript has been carefully re-checked and we have made the necessary changes according to your comments. We look forward to working with you and the reviewers to move this manuscript closer to publication in PLOS ONE.

The revisions in the manuscript are highlighted in yellow to facilitate the review process. Please also find below our point-by-point responses to all your comments. We hope that our explanations and corrections are satisfactory.

We look forward to hearing from you at your earliest convenience.

Sincerely,

Ayumu Ito, MD, PhD

Department of Obstetrics and Gynecology, Faculty of Medicine, Toho University

6-11-1, Omorinishi, Ota-ku, Tokyo, Japan

Phone: +813-3762-4151

Email: ayumu.itou@med.toho-u.ac.jp

---

## [Editor Report · Decision Letter 1]

9 Apr 2024

Endometrial congestion is the only hysteroscopic finding indicative of chronic endometritis

PONE-D-24-02159R1

Dear Dr. Ayumu,

We’re pleased to inform you that your manuscript has been judged scientifically suitable for publication and will be formally accepted for publication once it meets all outstanding technical requirements.

Kind regards,

Muhammad Salman Bashir, M.S.C

Academic Editor

PLOS ONE